# Developmental Language Disorder (DLD) in Young People Leaving Care in England: A Study Profiling the Language, Literacy and Communication Abilities of Young People Transitioning from Care to Independence

**DOI:** 10.3390/ijerph18084107

**Published:** 2021-04-13

**Authors:** Judy Clegg, Ellen Crawford, Sarah Spencer, Danielle Matthews

**Affiliations:** 1Division of Human Communication Sciences, Health Sciences School, University of Sheffield, Sheffield S10 2TS, UK; sarah.spencer@sheffield.ac.uk; 2Department of Psychology, University of Sheffield, Sheffield S1 2LT, UK; Ellen.crawford0@gmail.com (E.C.); Danielle.matthews@sheffield.ac.uk (D.M.)

**Keywords:** care leavers, developmental language disorder (DLD), language, communication, literacy, psycho-social, mental health

## Abstract

Research indicates children and young people in care have a high prevalence of Developmental Language Disorder (DLD) as part of a complex set of vulnerabilities. This study describes the profile of language, literacy and communication abilities of a cohort of care leavers. The language, literacy and communication abilities of 44 young people leaving care between the ages of 16 and 26 years were assessed using standardized measures. Demographic data about the young people was collected along with a survey to key staff to capture their perceptions and experiences of the language and communication abilities of these young people. Ninety percent of the care leavers’ language abilities were below average and over 60% met criteria for DLD in combination with literacy difficulties, developmental disorders and social, emotional and mental health difficulties (SEMH). The implications of unidentified DLD on the lives of young people leaving care is discussed. Earlier identification of DLD is advocated to enable services to intervene to facilitate more positive outcomes and life chances for this very vulnerable population.

## 1. Introduction 

### 1.1. Children and Young People in Care 

In 2018, there were 75,429 looked after children in England [1]. These children are taken into the care of their Local Authority and may live with foster parents, in a residential children’s home or in other residential settings such as schools and secure units. The number of children 16 years and older who are unaccompanied asylum-seeking young people (absent parenting) who are placed in care is increasing [1]. When children reach 16 years of age, they start to make the transition away from looked after care to independence. These young people continue to be looked after by the local authority but move into leaving care services where they continue to be supported up to 25 years of age. Key staff, often termed Personal Advisors (PAs) support young people in their transition to ensure they receive the financial, educational and other support they are entitled to. 

Children in looked after care are very vulnerable with poor life chances across educational attainment, employment, independence and social, emotional and mental health (SEMH) [2,3,4]. There are complex and multi-factorial issues operating for these children. Early development usually takes place in the context of psycho-social adversity with risks of attachment difficulties, late or under-identification of neuro-developmental disorders, difficult and traumatic experiences (abuse, neglect), and educational disadvantage [2]. Due to their complex needs, many of these children may have language, literacy and communication difficulties which are not identified and supported. It is not clear how these difficulties may contribute to their vulnerabilities, especially as they transition into adult life. 

### 1.2. Language, Literacy and Communication Profiles of Children and Young People in Care

Research investigating the language, literacy and communication profiles of children and young people in care is preliminary [5]. A recent study examined the language, literacy and mental health profiles of 26 adolescents with a mean age of 15 years in out-of-home care in Australia [5]. On a language measure, 92% of the sample scored below average with 62% of the sample scoring two or more standard deviations below the mean. Similar scores were identified on a reading and a self-report measure of discourse and pragmatic skills. Over 50% of the sample self-reported mental health difficulties and 23% self-reported at least one diagnosis of a neurodevelopmental disorder. Despite the high rate of DLD identified, only one participant self-reported receiving a diagnosis indicating speech, language and communication needs. The findings indicate a high prevalence of long standing and unidentified DLD. Although this study did not find clear associations between DLD and SEMH difficulties, the findings indicate the involvement of DLD in their poor life outcomes. In a USA sample of 70 adolescents entering a residential treatment setting, 75% were identified with language impairments [6]. In a Scottish study [7], residential staff completed the Children’s Communication Checklist-2 (CCC-2) [8] for 30 young people in residential care aged between 11 and 17 years. Nineteen of the 30 young people were identified with impairments; eight with impairments indicative of Autism Spectrum Disorders (ASD) and the remaining 11 participants with impairments across other aspects of speech, language and communication. Many children and young people in looked after care have experienced maltreatment [9,10]. A meta-analysis of 26 studies investigating language abilities in maltreated children confirmed an association between maltreatment and poor language skills [10]. Severe parental neglect is reported to impact more on language development than physical abuse [11,12,13]. Children who are neglected are more likely to stay in the home where language stimulation is poor, whereas children who are physically abused are usually removed from the parental home into a more verbally stimulating environment [12]. 

The term DLD refers to children who have significant developmental difficulties in their language and communication abilities. The term DLD has recently been revised and is used to refer to children who show difficulties across many modalities of language and communication (including speech). These difficulties cannot be best explained by intellectual disabilities, other neuro-developmental disorders, hearing loss or other neuro-biological reasons [14]. There is a substantial body of cross sectional and longitudinal clinical studies confirming high rates of SEMH in children and young people with DLD [15,16,17] and indeed the co-morbidity between DLD and SEMH is now acknowledged [14,18]. The DLD is usually identified as a primary impairment and the SEMH difficulties as a secondary consequence emerging later in development as the demands of growing older increase. However, in contrast to clinical studies, population cohort data is less clear about this association. Analysis of adolescents in two UK population studies [19] did not find relationships between child language ability and adolescent self-reported internalizing symptoms. Analysis of children from another UK population study identified some association between children’s early language and their social, emotional and behavioural functioning in the first six years of life [20]. It is important to note the association between language difficulties and SEMH do not consistently hold in population studies. This is likely due to the variation in measures used and a paucity in the amount and quality of clinical data available in these studies. However, given the many vulnerabilities of children in care, it is likely some of these children also present with DLD. Furthermore, the DLD may be overlooked due to their many other complex psychosocial needs. Alternatively, the DLD may be associated with their early development in the context of psycho-social deprivation. If children in care do have high rates of DLD then services need to identify and support these needs to facilitate more positive life chances.

This current study builds on previous research [5,6,7] by profiling the language, literacy and communication abilities of young people aged 16 years and over who are moving from care into independence. Given the age of these young adults, it is important to understand if and how standardised assessments align with their own perceptions of their language and communication abilities in addition to the perspectives of the professionals who support them. This has the potential to further our understanding about DLD in this population and how their needs are perceived and met. 

### 1.3. Current Study: Aims and Research Questions

The aims of the study were to (1) describe the profile of language, literacy and communication abilities of a cohort of care leavers aged 16 to 25 years; (2) determine if this profile of abilities is self-reported by the care leavers themselves and the key staff working with them and: (3) understand how these key staff perceive and experience the language and communication abilities of the young people they work with.

The study asked the following research questions: What is the language, literacy and communication profile as measured by standardised assessments of care leavers aged between 16 and 25 years? How do care leavers report their own language and communication abilities and does this align with the profile of language and communication ability described by the assessments? Can key staff working with care leavers describe their language and communication abilities and does this align with the profile of language and communication ability described by the assessments?How do key staff perceive and experience the language and communication abilities of care leavers? 

## 2. Materials and Methods

### 2.1. Study Design

This is a cross sectional study using standardised assessments to describe the profile of language, literacy and communication abilities of care leavers aged between 16 years to 26 years. The study employs an online survey to capture the perspectives and experiences of key staff about the language and communication abilities of the young people they work with. 

### 2.2. Participants

Participants were 44 young people aged between 16 and 26 years with a mean age of 20 years and 9 months (SD 2.20). Thirty were male and 14 were female. Participants were identified through opportunity and voluntary sampling from the council leaving care service in a large city in England. All young people in the service were eligible to participate (from an approximate total of 250 young people at the time of the study). Study information was sent to all the Personal Advisors (PAs) in the service to explain the study and the potential involvement of the young people they work with. PAs are professionals who play a key role in supporting young people with their transition to independence. PAs were asked to discuss the study with all their young people. Where young people indicated they would like to participate, the PA contacted the research team and an assessment was set up. Flyers about the study were also distributed through a venue where the service is based which functions as a drop in-centre for the young people. Interested young people were asked to make contact with the research team. Researchers also attended a young person led support group to tell young people about the study as well as distributing flyers to young people as they came into the venue. If a young person indicated an interest in participating in the study, they received an information sheet (accessible version) and an assessment was set up at a convenient date and time. PAs were also participants in the study. Twenty-three PAs completed an online survey to capture their perspectives and experiences of the language and communication abilities of the young people. 

### 2.3. Measures 

Participants completed assessments of language, literacy and communication. 

#### 2.3.1. The Clinical Evaluation of Language Fundamentals (CELF-5 UK)

The CELF-5 UK^19^ is an assessment of language standardised on a UK population aged 5 years to 21 years and 11 months. Four subtests of the CELF-5 UK were administered to form the composite Core Language Score (CLS) [21]. The core language subtests are as follows:

##### Formulating Sentences (FS)

This subtest measures the ability to construct grammatically and semantically correct sentences. The participant is presented with a picture and a spoken word and then asked to create a spoken sentence about the picture using the word. A score of two is given for a grammatically, semantically and syntactically correct sentence. A score of one is given where there is a complete sentence but there are one or two syntactic or semantic errors. A score of zero is given where there is an incomplete sentence, the sentence does not make sense or there are multiple errors. 

##### Recalling Sentences (RS)

This subtest measures listening, specifically verbal working memory. The participant is asked to immediately repeat a sentence spoken by the examiner. The sentences increase progressively in length and complexity. A score of three is gained where no errors are made and gradually decrease to zero, when four or more errors are made. 

##### Understanding Spoken Paragraphs (USP)

This subtest measures language understanding where participants answer questions relating to novel, factual information heard in a spoken paragraph. The examiner reads out written paragraphs of increasing length and complexity, and the participant is then asked to answer questions about the information they listened to. A score of 1 is awarded for each correct answer. 

##### Semantic Relationships (SR)

This subtest measures the ability to correctly understand sentences where a range of semantic relationships are expressed including comparison, locations and directions, time relationships, serial order and tenses. For each item, a maximum score of 1 is available. 

Raw scores were converted into scaled scores using the UK standardised norms of the CELF-5-UK enabling comparisons with UK age norm scores. A scaled score of 10 describes the mean score; scaled scores of 7 and 13 are one standard deviation (SD) below and above the mean. Scores lower than 7 are indicative of significant difficulties. The mean of the standard core raw score is 100. The CELF-5 UK has high reliability and validity.

#### 2.3.2. Test of Word Reading Efficiency 2 (TOWRE-2)

This [22] is a standardised assessment of single word reading consisting of the Sight Word Efficiency and Phonemic Decoding Efficiency subtests. The Sight Word Efficiency subtest asks the participant to read aloud English words from a list which progressively increase in phonemic complexity and length. This subtest is designed to assess sight-reading proficiency and an individual’s ability to accurately pronounce printed words. The Phonemic Decoding Efficiency subtest asks the participant to read aloud from a list of phonemically regular non-words which again increase in length and complexity. This subtest measures an individual’s ability to decode and accurately pronounce non-words. Across both subtests, a score of 1 is given for each correct response. The TOWRE-2 has a mean standard score of 100 with standard deviations of 15. A score of 85 or less indicates difficulties. The TOWRE-2 is standardised on a USA population aged 6 years to 24 years and 11 months. High reliability and validity are reported. 

#### 2.3.3. Communication Checklist Adult Self Report Scale (CC-SR) and Adult Report Scale (CC-A)

The Communication Checklist [23] is a 70-item questionnaire used to screen for language and communication difficulties in adults. It is a standardised assessment for adults between the ages of 17 and 79 years of age and takes 5 to 15 min to complete. There are two versions. The first is a self-report version (CC-SR) and the second is an adult report version (CC-A) [24]. The participants completed the CC-SR and the PAs to the participants completed the CC-A for their respective young person(s). 

In both the CC-SR and the CC-A, language and communication ability is rated across three domains of (1) Structural language (abilities in speech, grammar and vocabulary/word meanings); (2) Pragmatic skills (social communication behaviours) and; (3) Social engagement (non-verbal social communication behaviours and interests). A scaled score of 6 or less is taken as a cut off score indicating a difficulty. The participants were given the choice to complete the CC-SR independently or to have the researcher read out each statement and record their score. The majority of participants asked the researcher to read the statements aloud to them and record their score. The CC-A was completed by the participant’s PA. PAs completed this measure independently but were able to ask the researcher questions on certain items if needed. The PA may have completed this measure multiple times if they had more than one young person on their caseload participating in the study. 

#### 2.3.4. Demographic Questionnaire

Due to ethical issues, it was not feasible to access participants’ demographic data about the participants held by the service. A questionnaire was designed to capture data pertaining to current vocation (employment/studying/other), current living circumstances, number and type of care placements experienced, age entered and left care, diagnoses received, in receipt of a Statement of Special Educational Needs/Education, Care and Health Plan (EHCP), qualifications, physical health needs and criminal convictions. Where participants needed support, the researcher read out the questions and recorded their responses. 

#### 2.3.5. Survey

A short online survey was designed to capture the perspectives and experiences of PAs. The survey consisted of ten questions (see Appendix A) and 23 PAs from a total of 28 completed the survey. 

### 2.4. Procedure 

At the assessment, the participants received the information sheet (accessible version) again to read through or the researcher talked through the information. If the participant agreed to take part, he/she gave their written consent. Assessments were either conducted at the city council venue or externally at the young person’s home with the PA present. The assessments completed by the young people were the CELF-5 UK, TOWRE-2, CC-SR and the demographic questionnaire. All assessments were conducted in English and the order in which the assessments were completed was varied. The assessment session was approximately one hour in duration.

The majority of the assessments were carried out in the city council venue. It was not possible to use an individual room unless a PA was present (which was not always feasible). Therefore, some assessments may have been impacted by background noise. Participants received a £20 gift voucher for their time. PAs completed the CC-A in their own time and returned these to the researcher in person or by post. 

### 2.5. Analysis

The CELF-5 UK, TOWRE and CC-SR and CC-A data was examined for normality. Shapiro Wilks tests confirmed significance of non-normal distribution for the CELF-5UK data only. Therefore, the medians and inter-quartile ranges (IQR) are reported for the CELF-5 UK data. Means, standard deviations and ranges are reported for the TOWRE and CC-SR and CC-A. Due to the variation in the sample, a descriptive analysis was conducted. Non-parametric correlational analysis was also completed. SPSS-Version 21 was used to conduct all analyses. 

## 3. Results

### 3.1. Participant Demographic Data

Of the 44 participants, 30 were male and 14 were female with a mean age of 20 years and 9 months (SD 2.20) (range 16 to 26 years). It was noted that one participant was 26 years old and so beyond the age range of the Leaving Care Service. Table A1 (Appendix B) shows the self-report demographic data.

#### 3.1.1. Living Circumstances

Twenty-three (52.3%) participants (aged between 18 and 25 years) were living independently. Seven participants (15.9%) (aged 16 to 26 years) were in supported living making the transition into independent living. Six participants (13.6%) (aged 18 to 20 years) were still in foster care and 4 participants (9%) (aged 16 to 21 years) were categorised as ‘other’ (2 in hostel accommodation, 1 in a safe house and 1 as homeless). 

#### 3.1.2. Experience of Looked after Provision

Forty participants (91%) reported experience of foster care and 4 (9%) experience of foster and residential care. The mean age of entering care was 10 years and 9 months (range 2 years to 18 years). Of the 29 participants (66%) who had left care, the mean age of leaving was 17 years and 6 months (range 15 to 19 years). Fifteen participants (34.1%) reported still being in care and four (9%) were not sure/did not respond to this question. 

#### 3.1.3. English as an Additional Language (EAL)

Five participants (11.4%) reported speaking English as an Additional Language (EAL) These participants were aged between 18 and 21 years and had entered care when they came to England as unaccompanied asylum seekers at the age of 16 years or under. Four participants came to England from Eritrea and spoke Tigrinya as their first language. The fifth participant did not report which country he had travelled from and/or his first language. 

#### 3.1.4. Education and Vocational Circumstances 

At the time of the study, twenty-three participants (52.3%) (aged 16 to 26 years) were not in employment, education or training (NEET), nine participants (20.5%) were in paid employment (aged 18 to 25 years), ten (22.7%) in education (17 to 24 years) and two (4.5%) in volunteer roles (aged 20 to 23 years). Educational qualifications were categorised as highest educational qualification achieved ever. GCSEs were attained by 16 participants (36.3%), NVQs by 18 participants (40.9%), BTECs by 5 participants (11.4%) and A levels by 1 participant (2.3%). No participants in this sample had gained a higher education/University qualification. 

#### 3.1.5. Special Educational Needs 

Nineteen participants (43.2%) reported receipt of either a Statement of Special Education Need (SEN) and/or an Education and Health Care Plan (EHCP). Twelve of these 19 participants (27.3%) were unable to report why they had received additional education support. Seven participants (15.9%) did report reasons across learning difficulties (n = 3), social, emotional and mental health difficulties (n = 1), reading and/or dyslexia (n = 2), physical/mobility/hearing (n = 1). Fourteen participants (31.8%) reported receipt of a developmental disorder diagnosis ever across Attention Deficit Hyperactivity Disorder (ADHD) (n = 8), Autism Spectrum Disorder (ASD) (n = 5), learning difficulties (n = 4), dyslexia (n = 4) and dyspraxia (n = 1). 

#### 3.1.6. Health Circumstances 

Ten participants (22.7%) reported physical health needs including hearing difficulties (n = 3), epilepsy (n = 1), mobility needs (n = 4) and cardiac needs (n = 2). In terms of mental health difficulties, 29 participants (65.9%) reported mental health difficulties across depression (n = 13), anxiety (n = 15), bi-polar (n = 1), conduct disorders (n = 1), obsessive compulsive disorder (n = 2), borderline personality disorder (n = 2) and post-traumatic stress disorder (n = 1). 

#### 3.1.7. Experience of the Criminal Justice System 

Fifteen participants (34.1%) reported receipt of a criminal conviction. Four received a referral or community order and four received a custodial sentence. One was not sure and six participants chose not to disclose the type of sentence received. 

### 3.2. Language 

Complete CELF-5 UK scores were available for 42 of the 44 participants (Table 1. The median composite CELF-5 UK core language score was 67.00 (IQR 21; range 40–111). In total, 90.5% (n = 38) were in the below average category (within −1SD to −1.5SD) and 72.7% (n = 32) in the low to severe range (within −1.5SD to −2 SD and below). Only four participants (9.5%) were in the average/above average category. Median scaled scores on the CELF-5 UK subscales were slightly higher on Understanding Spoken Paragraphs and Semantic Relationships. According to the CATALISE definition of DLD, 27 of the 44 participants met criteria for DLD (CELF-5 UK scores −1.5 SD to −2 SD and below and excluding the 5 participants with EAL. 

### 3.3. Literacy

Thirty-nine participants completed the TOWRE-2 (see Table 2). The mean TOWRE-2 sight word efficiency scaled score was 84.69 (SD 17.91; range 55–118) and the mean TOWRE-2 phonemic efficiency scaled score was 89.90 (SD 18.53; range 57–123). The overall total TOWRE-2 scaled mean score was 86.62 (SD 18.45; range 54–118). Seventeen participants (38.6%) were categoriseBelow averagd as average and above average, six (13.6%) as below average and 16 (36.4%) in the poor/very poor categories. 

### 3.4. Communication Checklists 

#### 3.4.1. Communication Checklist—Self Report (CC-SR)

All 44 participants completed the CC-SR (see Table 3). The mean total scaled score was 7.17 (SD 9.15; range 2–44). The highest mean scaled score was for Pragmatic Skills and the lowest for Social Engagement. Seventeen participants (38.6%) gained a total mean scaled score of 6 or less indicating a communication difficulty. 

#### 3.4.2. Communication Checklist—Adult Report (CC-A)

Complete data was available for 21 of the 44 participants (see Table 4). The mean total CC-A scaled score was lower than the CC-SR. Similar to the CC-SR, the highest mean CC-A scaled score was for Pragmatic Skills and the lowest for Social Engagement. Thirteen participants (61.9% of n = 21) gained a total mean scaled score of six or less indicating a communication difficulty. 

#### 3.4.3. Comparison of CC-SR and CC-A Data

Comparison of the CC-SR and CC-A data was completed using the same 21 participants who had complete data for both scales (see Table 4). All 21 participants rated themselves higher than the PAs across Language Structure and Pragmatic Skills but not Social Engagement. On the CC-SR, six participants gained a total mean scaled score of 6 or less indicating a communication difficulty. On the CC-A, 13 participants received a total mean scaled score of 6 or less indicating a communication difficulty. Only four of the participants were identified with a communication difficulty by both the CC-SR and the CC-A. Spearman rank correlations between the CC-SR and the CC-A for each subscale and the total score only identified a significant correlation for the Pragmatic skills subscale (rs = 0.443; *p* = 0.45), There was a negative non-significant correlation for Language Structure (rs = −0.159; *p* = 0.492), a positive non-significant correlation for Social Engagement (rs = 0.041; *p* = 0.859) and a negative, non-significant correlation for the total score (rs = −0.134; *p* = 0.56).

### 3.5. Profile of Participants with CELF-5 UK Scores in the Very Low/Severe Range

The profile of the 26 participants with CELF-5 UK scores in the very low/severe range (−2 SD and below) was examined with respect to EAL, receipt of a Statement of SEN/EHCP, diagnoses of developmental disorders and mental health difficulties and scores of 6 or below on the CC-A and CC-SR (see Table A2 in Appendix C). Of the 26 participants, 11 self-reported receipt of a statement of SEN/EHCP, ten participants a diagnosis of developmental disorders and 11 participants, a diagnosis of a mental health difficulty. None of the self-reported diagnoses included DLD. Eleven confirmed self-reported language and communication difficulties on the CC-SR and seven were reported with language and communication difficulties by the PAs using the CC-A. There were 10 participants from this subsample of 26 participants with missing data for the CC-A. 

Importantly, the five participants who were unaccompanied asylum seekers with EAL were in this sample. These five participants with EAL were removed to describe the profile of language, literacy and communication of the 21 participants with CELF 5UK scores (−2 SD and below) (see Table 5). Descriptive comparison between these two subgroups showed the five participants with EAL scored lower across all four assessments. With only five participants in this subgroup, it was not feasible to run a statistical comparison. Interestingly, these five participants did not self-report any developmental disorders or mental health difficulties. However, this is probably explained by the recent move to the city and England as unaccompanied asylum seekers. The 21 mono-lingual participants continued to have CELF-5UK scores in the very low/severe range.

### 3.6. Survey Findings

The survey to the PAs was completed by 23 participants. The findings are presented for each of the questions. 

**Question** **1.**
*What is your staff role in the Children Looked After and Leaving Care Service, Sheffield City? (n = 23/23 responses)*


The majority of respondents were PAs (n = 20), with two as Senior PAs. One was a Senior Field Work Manager and the other two were from the Business Support Team who worked with care leavers in these roles. 

**Question** **2.**
*How long have you been in this role? (n = 23/23 responses)*


Six respondents had been in post for more than five years and ten respondents had been in post for less than a year. Of the remaining seven respondents, five had been in post between one and three years and two between three and five years. 

**Question** **3.**
*Do you experience difficulties in communicating effectively with the care leavers you work with? (n = 21/23 responses)*


Six respondents reported they do experience difficulties and seven reported they do not. One respondent was not sure, and seven selected other and responded with comments making reference to (1) care leavers changing their mobile numbers frequently; (2) variation across care leavers with some care leavers being effective communicators and others not; (3) English as an additional language and 4) avoidance of communication initiated by the care leavers. 

**Question** **4.**
*If there are care leavers with communication difficulties in your service, how are these communication difficulties documented? (n = 20 responses)*


The majority of respondents reported communication difficulties are documented in the Service Needs Assessment/Pathway Plans (n = 17) followed by SEN Statement/EHCP Plan (n = 9) and Case Files (n = 8). 

**Question** **5.**
*What types of communication difficulties do care leavers have? Respondents ranked 9 options from 1–9 where 1 represented the type of need most prevalent and 9 the type of need the least prevalent (n = 20/23 responses)*


The median ranks were calculated for the 9 options (see Table 6). Attention and listening to others was ranked as the most prevalent followed by understanding/processing what other say and being able to communicate with others in a socially appropriate way. Speech and hearing difficulties were the least prevalent. 

**Question** **6.**
*What support is available in your service for care leavers with communication difficulties? (n = 21/23 responses)*


No respondents reported being able to access a speech and language therapist (SLT) employed in the service whereas 11 respondents reported being able to access a SLT through another service. This was where care leavers were also involved in the Youth Justice Service which employs a SLT. One respondent was able to access support from staff who had received training about speech, language and communication needs (SLCN). Nine respondents reported not being able to access any support. 

**Question** **7.**
*Do you consider that having communication difficulties makes it more difficult for a care leaver to engage effectively in the service and their transition? (n = 22/23 responses).*


Twenty respondents reported yes and two reported no. 

**Question** **8.**
*If yes, please rank the following reasons from 1–5, where 1 represents the most common reason and 5, the least common reason (n = 16/23 responses).*


The median ranks were calculated for the 5 options (see Table 7). Giving inappropriate responses/behaving in a socially inappropriate way was the most common reason and finding it difficult to engage in verbally mediated interventions was the least common reason. The other 3 options were equally ranked. 

**Question** **9.**
*How effective do you consider communication between staff and care leavers is? (n = 20/23 responses)*


Communication between staff and care leavers is perceived as very effective by one respondent, effective by 12 respondents, medium effectiveness by six respondents and some effectiveness by one respondent. 

**Question** **10.**
*How could staff be supported to communicate more effectively with care leavers? (n = 23/23)*


Participating in training both to understand SLCN (n = 15) and to communicate more effectively with care leavers (n = 9) along with accessing speech and language therapy services (n = 14) and accessing resources to increase the accessibility of information for care leavers (n = 13) were all selected 

## 4. Discussion

The study conducted a descriptive analysis of the language, literacy and communication profiles of 44 young people leaving care, aged between 16 and 26 years. Young people rated their own language and communication abilities and PAs also rated the language and communication abilities of these young people. PAs completed a survey to understand their perspectives and experiences of the language and communication abilities of the young people they work with. 

### 4.1. Summary of Findings

The demographic data confirmed the poor psycho-social experiences and outcomes of young people in care [2,3,4]. For these young people leaving care, there were high rates of NEET, developmental disorders and SEMH. A high proportion of the young people showed language abilities in the below average range with 27 young people meeting the CATALISE definition of DLD. None of these 27 participants (61.3%) had ever received a label of DLD or diagnoses indicating speech, language and communication needs. Although, scores on the single word reading assessment were higher than the CELF-5 UK scores, literacy difficulties were also evident. This high prevalence of language difficulties was not supported by the CC-SR and CC-A. Although the CC-SR and CC-A scores were in the low range they did not meet the threshold for difficulty. The PAs did not identify the same profile of language and communication difficulties as the CELF-5 UK. 

The PAs were uncertain about developmental language and communication abilities and difficulties. There was agreement language and communication difficulties impede a young person’s transition and accessing appropriate support is challenging. Interestingly, if the young person was also involved in the youth justice service, they were able to access speech and language therapy services. Despite the variation in responses, there was confirmation language and communication difficulties make it harder for care leavers to engage in their transition to independence. Further training, access to speech and language therapy services and resources were all considered necessary. 

This study confirms the findings of the few previous studies identifying similar profiles of language, literacy and communication abilities and prevalence of DLD [5,6,7] in young people in care. The high rates of self-reported developmental disorders, SEMH and poor life outcomes identified is also similar to previous research [5]. It is concerning that 61.3% of these young people transitioning into independence meet criteria for DLD. The nature of the associations between the DLD and their psycho-social outcomes remains unknown. Children with DLD who are not in care are at risk of a range of negative psycho-social outcomes [15,16,17]. Mechanisms to explain this risk focus on the DLD as a primary impairment impacting on other aspects of development and attainment [15,16,17] in some way. For this cohort of young people transitioning out of care, it is not clear if the DLD is an unidentified primary impairment or a profile of DLD resulting from a complex and dynamic set of vulnerabilities or even both. Nevertheless, young people leaving care need to have their language, literacy and communication needs identified and supported. This includes young people with EAL who have recently arrived in England as unaccompanied asylum seekers. 

The findings are relevant to other vulnerable populations [25,26]. Young people in the criminal justice system (CJS) have a similar prevalence of DLD. In a sample of 145 first time young offenders with a mean age of 15 years, 60% met criteria for DLD and language difficulties were identified as a key predictor of their re-offending [25]. Compared to non-offenders matched on non-verbal intelligence, social disadvantage and years of education, 52 young offenders with a mean age of 16 years had a higher incidence of DLD [26]. Young people with experience of care are over-represented in the CJS [27]. Given both young offenders and young people leaving care have a high prevalence of DLD, language and communication difficulties must be addressed to decrease their vulnerabilities. Multi-agency support with access to speech and language therapy services has the potential to facilitate more positive life chances.

### 4.2. Methodological Limitations

There are several methodological limitations to highlight. The self-report demographic data and analysis confirms this sample is very heterogenous. Variation is evident across all the measures employed. The inclusion of participants who entered looked after provision as asylum seekers with EAL and the high rate of self report of Statements for SEN/EHCP, developmental disorders and mental health difficulties is further evidence of this. The sample of 44 participants (from a total number of approximate 250 care leavers in the service) is a small proportion of all the young people in the leaving care service at the time of the study and therefore may be biased. The study may have attracted those young people who were interested in receiving a gift voucher and/or actively seeking more support than other young people in the service. Furthermore, since the PAs knew the study was about language, literacy and communication, they may have been more likely to identify young people they considered could benefit from participating although measures were taken to prevent this. 

There are issues around the assessments and measures used. There is a dearth of language and communication assessments standardised for young people with potential developmental as opposed to acquired language and communication disorders. The CELF-5 UK is standardised up to 21 years and not 25 years. Although the TOWRE-2 is standardized to 25 years, it only provided a measure of single word reading and not more functional literacy skills. There is incomplete data, specifically for the PA completed CC-A assessment and some of the survey questions. It was challenging to ensure PAs completed the CC-A for their young people. For some young people, there were distractions in the venue where the assessments were completed which may have impacted on their scores. Indeed, many young people needed reassurance to complete the assessments. The CELF-5 UK employs a discontinuation rule where a young person has to continue attempting items before the stop rule is reached even though the young person is aware, they are failing. This is an uncomfortable experience and may have stopped young people engaging fully.

An important part of the study was to identify how young people perceive their own language and communication abilities and if this aligned with their CELF-5 UK scores. Some young people self-reported language and communication difficulties but there was no clear alignment between the self-reporting on the CC-SR with the PA completed CC-A and the standardised assessments. The CC-SR and CC-A did not correlate nor align with the CELF-5 UK. This is clearly an area for future research to consider how best to gain a more functional measure of language and communication ability from the perspective of the young person and those who work closely with him/her. 

The incomplete responses for some of the survey questions indicates the survey design not fully appreciating the limited knowledge and understanding of the PAs leading to potential mis-understanding and non-completion. The demographic data collected was via self-report as it was not possible to access participant data from the service due to ethical issues. The self-report demographic data may not be accurate. Indeed, several participants were unable to recall important historical information, e.g., reasons for having a Statement of SEN/EHCP plan. Nevertheless, the high proportion of young people identified with severe language and communication difficulties meeting criteria for DLD in this sample warrants further consideration.

## 5. Conclusions

Ninety percent of the care leavers had below average language abilities and over 60% met criteria for DLD in combination with literacy difficulties, developmental disorders and SEMH. Earlier identification of DLD is advocated to enable services to intervene to facilitate more positive outcomes and life chances for this very vulnerable population. All children and young people in care should be able to access speech and language therapy services to identify their language and communication needs and to support these effectively through childhood and into adult life. 

## Figures and Tables

**Table 1 ijerph-18-04107-t001:** CELF-5 UK Core language median and mean subscale scores, standard deviations, ranges, and categories of ability.

	Number of Participants (n = 42)	Test Scaled Score	Category of Ability	Relationship to Mean
Core Language Score (CLS) Median 67.00 (IQR 21) Range 40–111	1 (2.4%)		Above average	+1 SD and above
	3 (7.1%)		Average	Within + or −1 SD
	6 (14.3%)		Marginal/below average/mild	Within −1 to 1.5 SD
	6 (14.3%)		Low range/moderate	Within −1.5 to −2 SD
	26 (61.9%)		Very low range/severe	−2 SD and below
CELF-Formulating Sentences (FS) Median scaled score 5.00 (IQR 5) Range 1–13	2 (4.8%)	13+ and above	Above average	+1 SD and above
	7 (16.7%)	8 to 12	Average	Within and or −1 SD
	33 (75.0%)	7 and below	Below average	−1 SD and below
CELF-Recalling Sentences (RS) Median scaled score 5.00 (IQR 4) Range 1–12	0	13+ and above	Above average	+1 SD and above
	7 (16.7%)	8 to 12	Average	Within and or −1 SD
	35 (83.3%)	7 and below	Below average	−1 SD and below
CELF-Understanding Spoken Paragraphs (USP) Median scaled score 3.00 (IQR 5) Range 1–16	1 (2.4%)	13+ and above	Above average	+1 SD and above
	4 (9.5%)	8 to 12	Average	Within and or −1 SD
	37 (88.1%)	7 and below	Below average	−1 SD and below
CELF-5 Semantic Relationships (SR) Median scaled score 4.00 (IQR 4) Range 1–13	2 (4.8%)	13+ and above	Above average	+1 SD and above
	3 (7.1%)	8 to 12	Average	Within and or −1 SD
	37 (88.1)	7 and below	e	−1 SD and below

**Table 2 ijerph-18-04107-t002:** TOWRE-2 single word reading mean scores, standard deviations, ranges and percentiles.

TOWRE-2	Raw Scores Mean (S.D) Range	Percentiles	Scaled Scores Mean (S.D) Range	
Sight Word (n = 39)	70.59 (22.98) 20–104	31.12 (27.27) 1–89	84.69 (17.91) 55–118	
Phonemic Decoding (n = 39)	39.15 (18.0) 7–63	40.32 (31/08) 1–94	89.90 (18.53) 57–123	
Sum of scaled scores (n = 39)			174.67 (35.0) 112–234	
Overall total scaled score (n = 39)			86.62 (18.45) 54–118	Above average 3 (6.8%) Average 14 (31.8%) Below average 6 (13.6%) Poor 6 (3.6%) Very poor 10 (22.7%)

**Table 3 ijerph-18-04107-t003:** Communication Checklist—Self Report (CC-SR) and Adult Report (CC-A) mean scores, standard deviations, ranges and percentiles.

CC-SR	Raw Score	Scaled Score	Z Score	Percentile
Language Structure (n = 44)	17.89 (10.74) 1–45	7.18 (3.6) 1–15	−1.17 (3.11) −19.9 to 1.8	31.01 (29.03) 0.1 to 96.4
Pragmatic Skills (n = 44)	19.61 (13.48) 0–47	7.50 (3.8) 0–17	−0.71 (1.34) −4.3 to 2.5	34.93 (30.42) 0.3 to 99.4
Social Engagement (n = 44)	28.27 (10.52) 5–46	6.84 (3.7) 0–15	−0.88 (1.21) −3.5 to 1.7	30.50 (28.25) 0.3 to 95.5
CC-SR Mean totals	21.92 (28.19) 8–120	7.17 (9.15) 2–44	−1.98 (1.45) −19.9 to −2.5	32.14 0.3 to 99.4
**CC-A**				
Language Structure (n−21)	10.33 (10.38) 0–43	6.62 (3.7) 0–14	−1.40 (1.10) −4.0 to 0.1	12.93 (14.62) 0–42
Pragmatic Skills (n = 21)	11.90 (12.69) 0–50	7.71 (4.4) 0–14	−1.09 (1.30) −4.0 to 0.4	8.99 (13.5) 0–50
Social Engagement (n = 21)	31.48 (17.11) 2–64	4.14 (3.8) 0–10	−2.26 (1.45) −4.0 to 0. 2	6.14 (11.03) 0–42
CC-A Mean totals	17.90 (34.80) 3–157	6.16 (10.10) 1–38	−1.79 (1.32) −4.0 to 0.4	5.51 (6.12) 0–18

**Table 4 ijerph-18-04107-t004:** Descriptive comparisons between CC-SR and CC-A raw scores for 21 participants.

	Language Structure		Pragmatic Skills		Social Engagement		Mean Totals	
	CC-SR	CC-A	CC-SR	CC-A	CC-SR	CC-A	CC-SR	CC-A
Raw Score SD Range	16.29 (8.22) 8–38	10.33 (10.38) 0–43	16.67 (10.83) 17–48	11.90 * (12.69) 0–50	29.29 (9.08) 17–48	31.48 (7.11) 2–64	62.24 (21.11) 38–103	53.71 (34.80) 3–157

* Correlation significant at 0.05 (2 tailed).

**Table 5 ijerph-18-04107-t005:** Profile of the participants with very low/severe CELF-5 UK scores who are monolingual (n = 21) compared to participants with EAL (n = 5).

	CELF-5 UK Core Language Score	TOWRE Total Score	CC-SR	CC-A
Monolingual participants (n = 21)	median 61.00 IQR 32 range 40–72 n = 21	mean 82.39 (SD 17.57) range 54–110 n = 18	mean 7.40 (2.98) range 2.6–12.0 n = 21	mean 6.55 (SD 2.77) range 3–11.3 n = 9
Participants with EAL (n = 5)	median 54.00 IQR 27 Range 40–67 n = 5	mean 76.80 (SD 11.82) range 57–88 n = 5	mean 6.60 (SD 2.40) range 2–9.66 n = 5	mean 8.33 (SD 3.18) range 5–11.33 n = 3

**Table 6 ijerph-18-04107-t006:** Survey: Question 5. What types of communication difficulties do care leavers have? Respondents ranked 9 options from 1–9 where 1 represented the type of need most prevalent and 9, the type of need the least prevalent (n = 20/23 responses).

Question 5 Response Options	Median Rank
Attending and listening to others (n = 20)	1.00
Understanding/processing what others say (n = 20)	2.00
Being able to communicate with others in a socially appropriate way (n = 20)	2.00
Reading and writing difficulties (n = 20)	2.50
Using appropriate vocabulary in their talking (n = 20)	3.00
Using appropriate grammar in their talking (n = 20)	4.00
Speech difficulties where speech is difficult to understand (n = 20)	5.00
Stammering/stuttering (n = 20)	6.50
Hearing difficulties (n = 20)	6.50

**Table 7 ijerph-18-04107-t007:** Survey: Question 8. If yes, please rank the following reasons from 1–5, where 1 represents the most common reason and 5, the least common reason (n = 16/23 responses).

Question 8 Response Options	Median Rank
The young person is more likely to give inappropriate communicative responses and/or behave in a socially inappropriate way which can have negative consequences for the young person (n = 16)	2.00
Staff find it hard to communicate with the young person which impacts on the overall management of the young person (n = 18)	3.00
The young person finds it difficult to express/regulate their emotions in respect to leaving care which may impede communication (n = 19)	3.00
The young person finds it difficult to access and understand information about the service offered (n = 17).	3.00
The young person finds it difficult to engage in verbally mediated interventions offered to support him/her (n = 18)	4.00

## Data Availability

This has been excluded.

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
