# Peer review of "Developmental Language Disorder (DLD) in Young People Leaving Care in England: A Study Profiling the Language, Literacy and Communication Abilities of Young People Transitioning from Care to Independence"

_ijerph, 2021, doi:10.3390/ijerph18084107_

Round 1

Reviewer 1 Report

This study is interesting in that the manuscript covers the very specific area which is rarely examined, that is, the lives of young people leaving care in terms of their language, literacy, and communication abilities. Thus, this study would greatly contribute to the field.

However, before that, this paper needs some revisions and improvements as follows:

First, I guess the context seems highly specific; thus I believe the more detailed explanation of the context of the study would be required for international readers to understand the importance and relevance of the results of the study. Otherwise, this study would be useful only for a certain group of researchers in the relevant field.

In addition, this study is conduced within the U.K. context; therefore, I guess it would be better for the authors to add ‘in England’ at the end of the title for readers to understand the field of this research.

Furthermore, in the discussion section, the authors also mention that the findings of the study confirms those of previous studies. Thus, it would be better to provide some findings that can distinguish this study from previous studies.

Also, the practical and hands-on implications would be desirable for the study to contribute to the relevant field.

Finally, there are some stylistic errors (e.g., spacing, placing a period etc.) throughout the paper, so please check the whole paper thoroughly.

Thank you.

Author Response

Please see the attachment where we have responded to each point raised by the reviewers. 

Reviewer 2 Report

The paper „Developmental Language Disorder (DLD) in young people leaving care: a study profiling the language, literacy and communication abilities of young people transitioning from care to independence” deals with very important and interesting problem. However, the paper (especially results section) should be rewritten before publication.

  • The most important limitation of this study is sample size – only 44 young people and 23 PA were involved in this study. With such small sample size the only some description of the problem is possible.
  • The results are displayed in so many small tables and later most of these data are rewritten in text once again. The tables should be combined and the most important results described in the main text, not every single number from the table. Some tables (e.g. tab. 10) should be moved to Appendix to make the results of the paper more clear.
  • The results for some scales are not normally distributed – to describe such variables median and quartiles should be used rather than mean value and SD.

Specific comments:

  • The age range of participants is 16-26 not 16-25 (see abstract, participant section, etc.)
  • In the Statistical analysis paragraph the more details of the data presentation should be displayed.
  • Comparison of CC-SR and CC-AR should be based on analysis of differences between the participant and AR rather than comparison of mean values for groups. It is very important to know if the scores are similar for the specific person, not for the group.
  • Tables 8 and 9 should be rewritten – for example median rank for each option should be displayed. These tables are very difficult to read now.
  • In the discussion – there is no need to repeat all the results. The most important findings should be repeated. There is a lack of comparison of these results with other populations. There is a lot numbers to describe scales, but it would be nice to see how these results corresponds to results from other population groups in similar age.
  • The sentence “The implications of unidentified DLD on the lives of children and young people in care are discussed” should be removed from Conclusions. This is not conclusion from this study.

Author Response

Please see the attachment were we have responded to the points raised by the reviewers. 

Round 2

Reviewer 1 Report

This version of the manuscript has been greatly improved by the integration of most of the first set of comments and suggestions. However, there are still some stylistic errors in the revised manuscript. Thus, it would be required for the authors to check the manuscript one more time.   

Author Response

Dear Editor

Please find attached the detailed response to the Academic Editor's recommendations. 

Many thanks

Judy
